# Hydrodynamics of the Vadose Zone of a Layered Soil Column

**Ioannis Batsilas, Anastasia Angelaki * and Iraklis Chalkidis**

Laboratory of Agricultural Hydraulics, Department of Agriculture Crop Production & Rural Environment, University of Thessaly, Fytokou Street, N. Ionia, 38446 Vólos, Greece
* Correspondence: anaggel@uth.gr

**Abstract:** Getting into the heart of the water movement into the vadose zone is essential due to the direct impact on the aquifer recharge, the flood hazards, the irrigation planning and the water resources management in general. Since soil profiles in nature appear in layers, the present study accomplishes a deep investigation of the water's motion through soil layers with different hydraulic properties. A series of experiments were conducted in the laboratory where infiltration, tension, soil moisture and hydraulic conductivity data were collected and analyzed. In particular, a transparent column was filled with two soils (loamy sand over sand), and TDR probes, along with ceramic capsules connected to pressure transducers, were set to the column. Using the experimental data and the unsaturated zone modeling, hydraulic parameters were obtained, along with water motion simulation and prediction. An investigation into the drainage, imbibition, infiltration, soil water characteristic curves and, in general, the hydrodynamics of the vadose zone of the soil layers has been achieved. The results of the current study suggest a method to estimate the crucial hydraulic parameters that are involved in the soil-water interaction and have an impact on infiltration, runoff, aquifer horizon recharge, water management and water saving.

**Keywords:** hydraulic parameters; hydraulic diffusivity; specific water capacity; sorptivity; hydraulic conductivity; retention curve; imbibition curve; soil water characteristic curve; infiltration; drainage

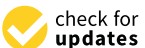



## 1. Introduction

The expansion of human activities has caused a spread of pollution, and nowadays, acid rain, hazardous chemical waste, fertilizers, insecticides, heavy metals, radioactive waste, sewage waste and solid waste are the most serious threats to the quality and quantity of groundwater and surface water [1]. Water movement through the soil has a great impact on aquifer recharge, groundwater level, water quality, runoff and irrigation planning [2,3]. In addition, it affects all the irrigation factors, such as irrigation dose, frequency, intensity etc. and therefore, water management and saving. In the last years, the climate crisis is constantly interacting with the soil-water complex affecting the hydrodynamics of the vadose zone. As the situation in several areas of the world is reaching crisis levels, the water demands exceed the available reserves [4,5]. Complete knowledge of the natural water-soil complex is hard to be achieved, so an approach system incorporates the development and validation of the simulation models, which can help scientists gain a better understanding of the complex and the interactive processes that affect water movement.

In nature, stratified soil profiles are very common; thus, layered soils are dominant [6]. Soil stratification affects the vertical water movement toward deeper levels [7]. Since vertical movement directly affects the infiltration process, along with chemical, pollutant movement and hydrological process in general, it is crucial to deeply investigate the phenomenon [8]. Chen et al., 2020 [9] researched the impact of the length of the layers along with the layer texture on crop production in an arid area. Hydrodynamics, along with texture effects and solute movement, can be investigated in situ and in the laboratory [10–12]. Water movement, along with solute transport, is related to soil texture; thus, water movement is easier through light-textured than fine-textured soils [13,14]. At coarse-textured soils,

as the soil dries, hydraulic conductivity decreases sharply [2]; hence, the presence of a soil layer with lesser hydraulic conductivity over the light-textured layer works to the direction of holding moisture at the upper layer due to capillary forces [15], while the under layer could prevent upward salt movement into the topsoil [16]. Putting the layers upside down can also increase the water-holding capacity if the hydraulic conductivity of the fine-textured layer is less than that of the coarse texture layer. Therefore, layered soils could favor vegetation establishment and increase yield under conditions of high evaporation and salinity [17]. Sun et al., 2021 [6] indicated that the water-holding capacity, storage, and infiltration process of sands could be affected by the presence of layers of medium-textured soils. Soil moisture depends on the soil texture configurations, the thickness of the layers, along with the silt and clay ratios. Generally, the soil layer with the lesser hydraulic conductivity controls the infiltration process in both cases (fine over coarse layer and upside down) [18]. Hu et al., 2021 [19] showed that the position and the thickness of the soil layers have a major impact on the infiltration characteristics. In addition, the ability of soil to hold water depends on many factors, such as porosity, hydraulic conductivity, sorptivity etc., and could be affected by natural hazards and pollution [12,15,20–24]. Hill and Parlange [25] proved that when the hydraulic conductivity of the lower layer is greater than the hydraulic conductivity of the upper layer, the soil moisture front is unstable. When the upper layer comes to saturation, the infiltration process starts at the lower layer. Cylindric fingers of water show up during the infiltration process of the lower soil layer when the hydraulic conductivity ratio is bigger than 20 [8]. Water motion in the unsaturated zone is described by Philip's [26–31] semi-analytic solution, which is given in the form of time series and came out from the Equation of water movement in homogeneous soil when we have flooding conditions on the surface. The equation of Philip is given below:

$$I(t) = K_i t + \sum_{m=1}^{m} S_m t^{m/2} \tag{1}$$

where $K_i$ is the hydraulic conductivity that corresponds to the initial soil moisture $\theta_i$, and $S_m$ is a series of coefficients that are calculated as functions of the soil characteristics and the initial and boundary conditions of infiltration. For $m = 1$, coefficient $S_m$ is called sorptivity [29,32], and it is valid for early times. Sorptivity is also given by Equation (2) which came out from soil moisture profiles, where $K_s$ is saturated hydraulic conductivity, $\theta_i$ is initial water content, $\theta_s$ is the boundary condition applied, $H_0$ is the load on the soil surface, and $H_f$ is the suction at the wet front.

$$S^2 = 2K_s(\theta_s - \theta_i)(H_0 - H_f) \tag{2}$$

Sorptivity can also be calculated from Vauclin and Havercamp's [33] Equation:

$$S^2 = 2 \int_{\theta_i}^{\theta_s} \theta D(\theta) d\theta \tag{3}$$

where $\theta$ is soil moisture, while $D(\theta)$ is diffusivity. Diffusivity is a hydraulic parameter given below:

$$D(\theta) = -\frac{K(\theta)}{C(\theta)} \tag{4}$$

where:

$$K(\theta) = K_s \left(\frac{\theta - \theta_r}{\theta_s - \theta_r}\right)^{1/2} \left\{1 - \left[1 - \left(\frac{\theta - \theta_r}{\theta_s - \theta_r}\right)^{1/m}\right]^m\right\}^2 \tag{5}$$

is hydraulic conductivity at the vadose zone and $C(\theta)$ is the specific water capacity given by the derivative:

$$C(\theta) = \frac{d\theta}{dh} \tag{6}$$

Hence, specific water capacity can be calculated as follows:

$$\theta = \theta_r + \frac{\theta_s - \theta_r}{[1 + (\alpha h)^n]^m} \Rightarrow \frac{d\theta}{dh} = -\frac{(\theta_s - \theta_r)\{[1 + (\alpha h)^n]^m\}'}{\{[1 + (\alpha h)^n]^m\}^2} \Rightarrow$$

$$\frac{d\theta}{dh} = -\frac{(\theta_s - \theta_r)m[1 + (\alpha h)^n]^{m-1}n(\alpha h)^{n-1}\alpha}{\{[1 + (\alpha h)^n]^m\}^2} \Rightarrow$$

$$\frac{d\theta}{dh} = -\frac{m\frac{1}{1-m}\alpha(\theta_s - \theta_r)(\alpha h)^{n-1}[1 + (\alpha h)^n]^{-1}}{[1 + (\alpha h)^n]^m} \Rightarrow$$

$$\frac{d\theta}{dh} = -\frac{m\alpha(\theta_s - \theta_r)(\alpha h)^{m/1-m}[1 + (\alpha h)^n]^{-1-m}}{1-m} = C(\theta) \tag{7}$$

Regarding the water movement simulation, Green & Ampt [34] proposed a model (GA model, Equation (8)), assuming that initial soil moisture is constant in a homogeneous soil column and the moisture profile is "piston-type", which indicates that the reduction of water content is sharp, under $H_f$ pressure head.

$$K_s t = I - \frac{S^2}{2K_s}\ln\left(1 + \frac{2K_s}{S^2}I\right) \tag{8}$$

$K_s$ is the saturated hydraulic conductivity, $S$ is the sorptivity, $I$ is cumulative infiltration, and $t$ is time. Parlange [35–39] also presented a model (P model) for the prediction of infiltration (Equation (10)) from Richard's Equation (9) [40], valid for one-dimensional water movement:

$$\frac{\partial \theta}{\partial t} = \frac{\partial}{\partial z}\left(D\frac{\partial \theta}{\partial z}\right) - \frac{\partial K}{\partial z} \tag{9}$$

$$K_s t = I + \frac{S^2}{2K_s}\left[\exp\left(-\frac{2K_s}{S^2}I\right) - 1\right] \tag{10}$$

The prediction model of Mualem & van Genucten [41,42] involves soil moisture and suction of water into the soil pores and is given by Equation (11). The $h(\theta)$ curve is called the soil water characteristic curve (SWCC).

$$\Theta = \frac{\theta - \theta_r}{\theta_s - \theta_r} = \frac{1}{[1 + (ah)^n]^m} \tag{11}$$

where $\theta$ is the water content, h is the suction of water into the soil's pores, $\theta_r$ is the residual water content, and $\theta_s$ is the saturated water content. The hydraulic parameters a, m, and n indicate the shape and curvature of the SWCC curve and *m* and *n* are related to each other as follows:

$$m = 1 - \frac{1}{n}, \ 0 < m < 1 \tag{12}$$

The complete knowledge of the mechanism of infiltration through soil layers comprises an important and necessary tool for decoding water movement. In addition, the estimation of hydraulic parameters that have a huge impact on water movement is essential as it can lead to further water-saving practices, such as irrigation planning and rational water management. It seems that there is a scientific gap in the determination of hydraulic parameters that have a major impact on water motion into unsaturated layered soils. The main scope of this study is to research deep on the water movement at the vadose zone of a layered soil column. For this purpose, a series of experiments were conducted in the laboratory, investigating the drainage, imbibition, infiltration, and in general, the hydrodynamics of the water movement. From the methods presented, crucial hydraulic parameters were determined, and water movement modeling was achieved for the soils under research.

## 2. Materials and Methods

The experiments of this study were performed in the Laboratory of Agricultural Hydraulics of the Department of Agriculture Crop Production & Rural Environment of the University of Thessaly. Two soil samples (loamy sand (LS-layer) and sand (S-layer)) well-dried at 105 °C and sieved with a classified device of sieves were used to fill a 6 cm diameter acrylic column at 2 layers. Bouyoucos hydrometer was used to determine the particle size distribution [43]. The 2 soil samples were placed into the column as follows: the LS-layer above S-layer. The height of the lower layer (S-layer) was 48 cm, and that of the upper layer (LS-layer) was 45 cm. To avoid possible porous media leaking, geotextile was placed at the bottom of the column. The geotextile had hydraulic conductivity at saturation greater than that of the soil samples in order not to significantly influence the water movement. The soil samples were packed uniformly, using a specific method, where soil passed through two sieves placed into a tube, which was used to fill the column. Attention was paid to keeping the distance between the lower part of the tube and the soil surface constant. The homogeneity of the soil column was checked at saturation, indicating homogeneity of porosity. In order to obtain soil moisture at various depths, TDR waveguides were placed at 7 cm, 23.5 cm, 33.5 cm, 48.5 cm, and 58.5 cm below the soil surface. Ponded irrigation simulation was achieved by applying a constant load of 2 mm water on the surface of the upper layer. Two volumetric tubes were used to measure the incoming water volumes, 1 was used to pour water on the soil surface, and the other one was used as an outflow tube that helped to keep the water load constant on the soil surface (Figure 1). Soil water tension was measured by placing 4 ceramic capsules (CC) in the column and connecting them to 4 pressure transducers (PT) respectively. So, each layer was supported with 2 CCs. The CCs were placed at 25 cm, 36.5 cm, 52.5 cm, and 63.5 cm from the soil surface (Figure 2). The system CC-PT was calibrated before use. For this purpose, a transparent column was filled with water, and the values of voltage at different tensions were noted (Figure 3). Then the calibration curves were figured and approximated with linear Equations (Figure S1). The factors of the linear Equations were inserted into the system's software. Soil moisture at saturation was priorly measured in the laboratory, and at the same time, saturated hydraulic conductivity ($K_s$) was measured for each layer using the constant head method experimental setup. The first drainage was achieved gradually by placing the Mariotte bottle in four positions (stages). At the beginning (1st stage), the Mariotte bottle was placed 25 cm from the soil surface, where it remained for 20,715 min. Then, in the 2nd stage, we put the Mariotte bottle at the interface of the layers (44.5 cm from the soil surface) for 22,318 min. In the 3rd stage, the Mariotte bottle was placed 79.5 cm from the soil surface for 8768 min, and during the 4th stage, it was placed 84.5 cm from the soil surface for 9686 min. The experiment lasted a total of 61,487 min (1025 h). After drainage, the 2nd imbibition followed, which was achieved in 4 stages. in the 1st stage, the Mariotte bottle was placed −21 cm from the 4th CC; in the 2nd stage, it was placed at the interface (−8 cm from the 3rd CC); in the 3rd stage it was placed at the height of the 1st CC (−11.5 cm from the 2nd CC), and at the 4th stage it was placed at a height equal to the soil surface (−25 cm from the 1st CC).

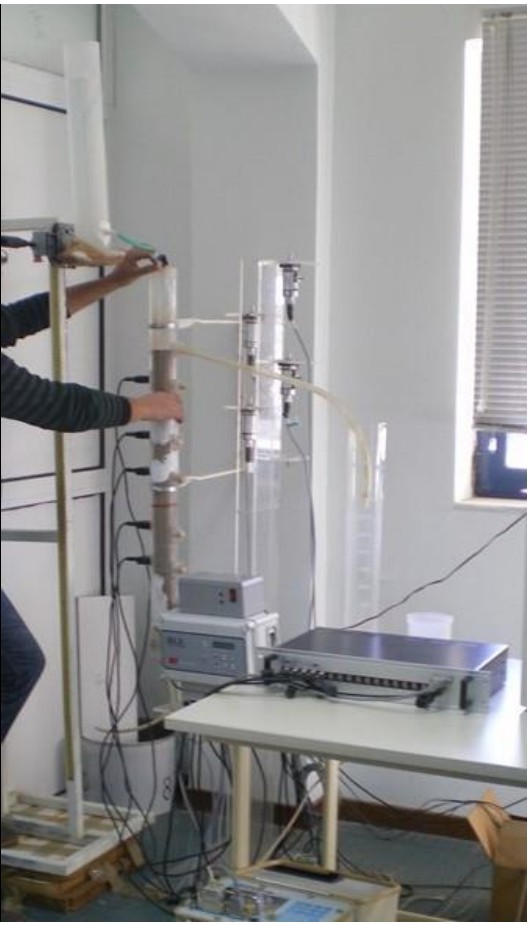

**Figure 1.** Infiltration under ponded conditions experimental setup.

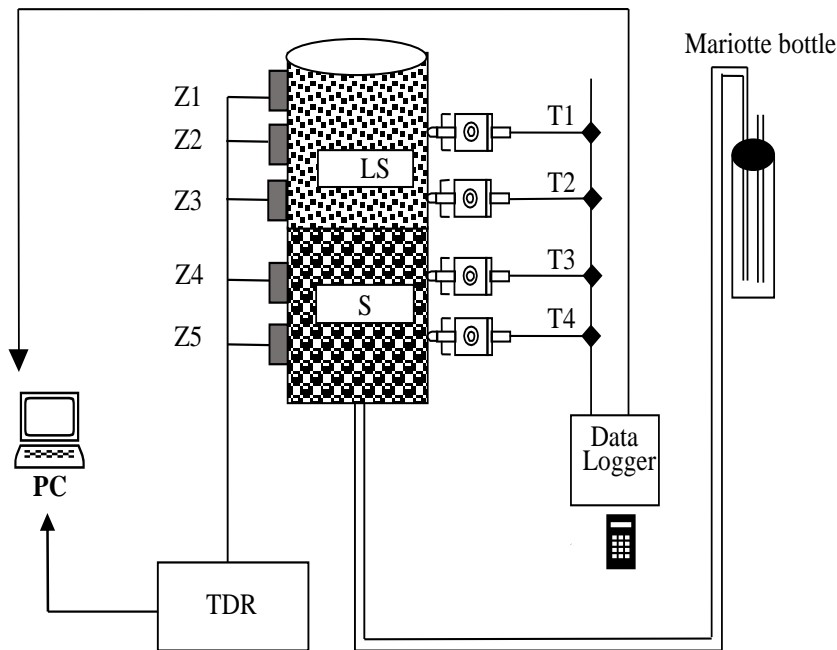

**Figure 2.** Experimental setup for the pressure head–soil moisture research.

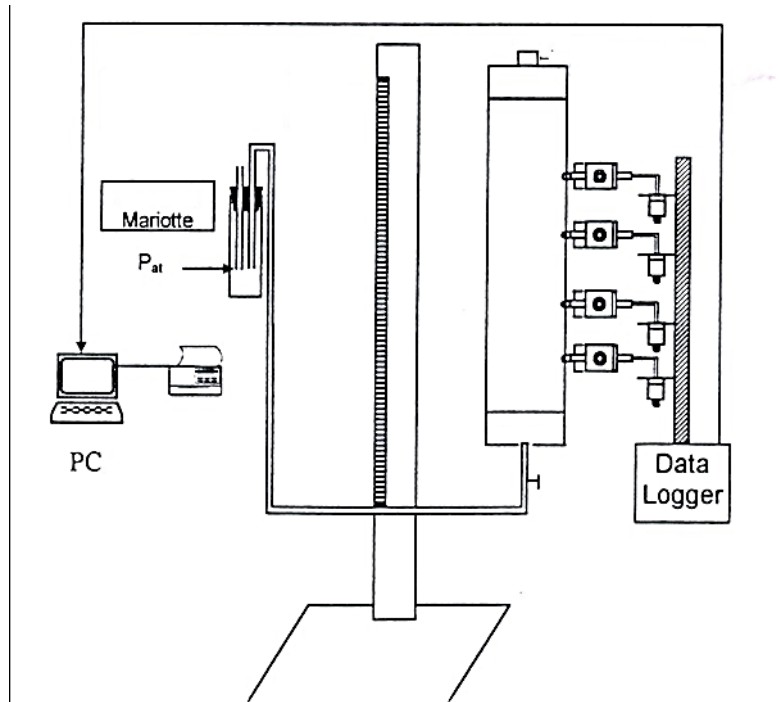

**Figure 3.** Experimental setup for the calibration of the P.T.

## 3. Results

The particle size distribution resulted that the upper layer consisted of 79% sand, 9% silt and 12% clay and was characterized using the USDA Soil Texture Triangle as loamy sand (LS-layer), while the under layer consisted of 94% sand, 2% silt, and 4% clay and characterized as sand (S-layer). Hydraulic conductivity and soil moisture at saturation ($K_s$ and $\theta_s$, respectively) for each layer were found:

Loamy sand (LS-layer): $K_s = 0.170$ cm min$^{-1}$, $\theta_s = 0.38$
Sand (S-layer): $K_s = 1.490$ cm/min$^{-1}$, $\theta_s = 0.26$

### 3.1. Infiltration under Ponded Conditions

Figure 4 presents the soil moisture values versus time at different depths, while the soil moisture profiles (water content vs. depth) are presented in Figure 5. Figure 4 shows that after about 80 min for the upper layer and 110 min for the underlayer, all soil moisture sensors showed stabilized values of soil moisture. In Figure 6, the incoming volumes of water vs. time are presented, and within the same figure, the volumetric water content that came out from the integration of the moisture profiles $\theta(z)$ is presented.

Sorptivity (S) of the upper layer (LS-layer) was calculated from Equation (1), for $m = 1$ and found:

$S_{LS} = 1.0607$ cm min$^{-0.5}$

At late infiltration times, the derivative $dI/dt$ was calculated, and the result was:

$dI/dt = 0.177$ cm min$^{-1}$.

Figure 7 shows the approximation of two well-known infiltration models (GA model and P model) with the experimental points of the upper layer. The relative mean squared error (Rmse) between the experimental points of the GA model was 0.11, while the relative mean squared error (Rmse) between the experimental points and the P model was 0.04.

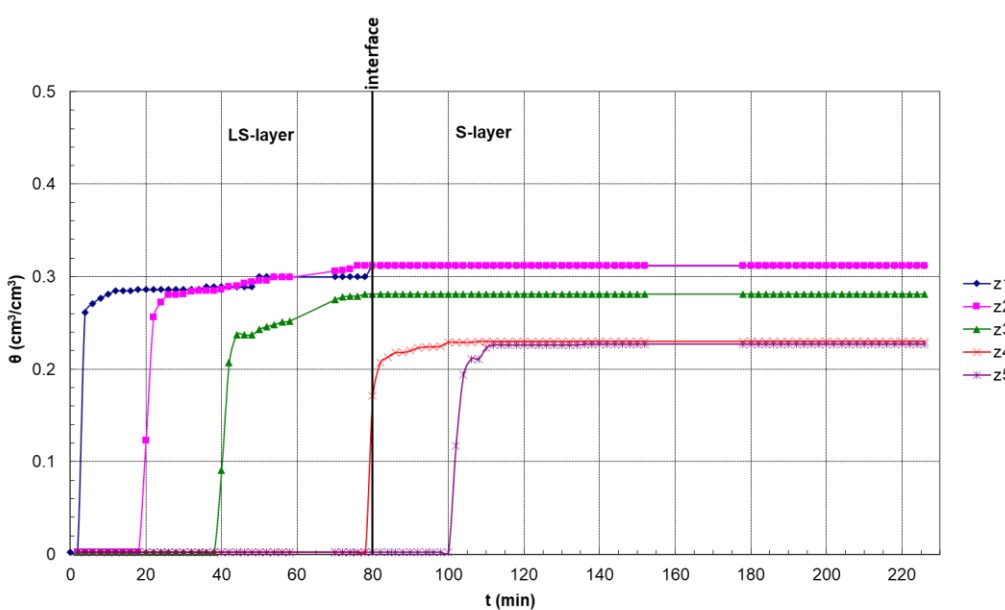

**Figure 4.** Soil moisture versus time at different depths.

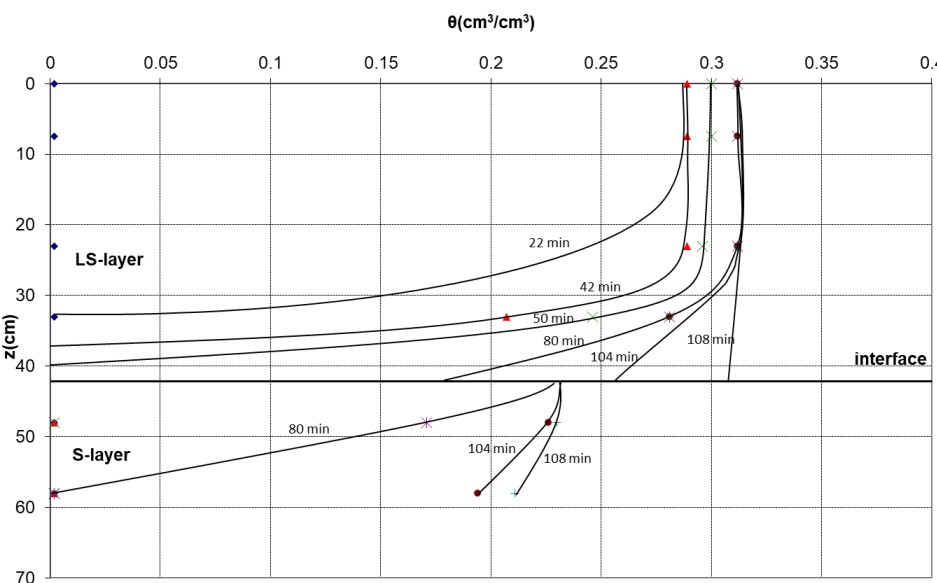

**Figure 5.** Soil moisture profiles (soil moisture versus depth).

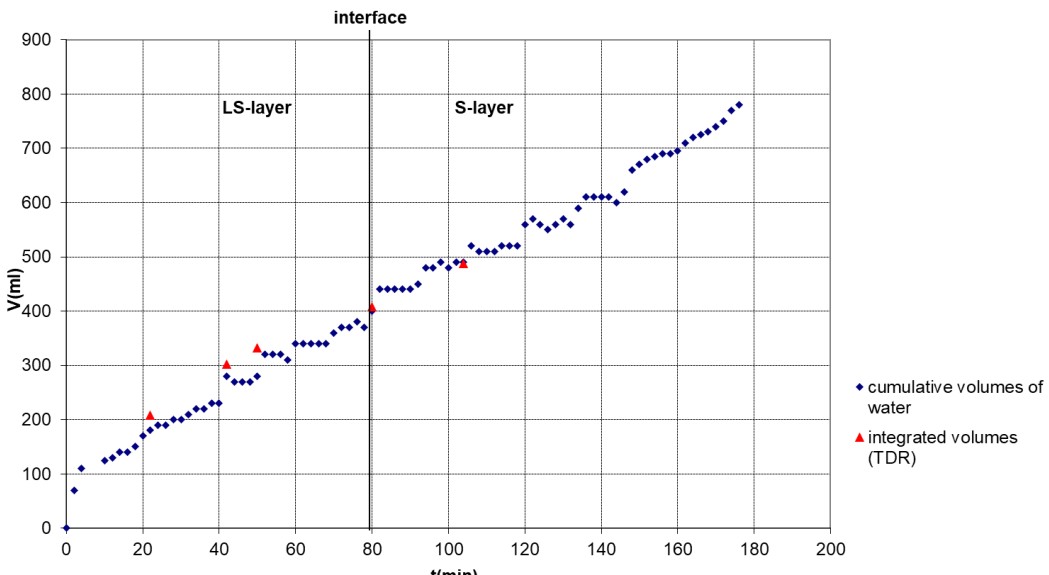

**Figure 6.** Infiltrated volumes of water and volumes from moisture profile integration.

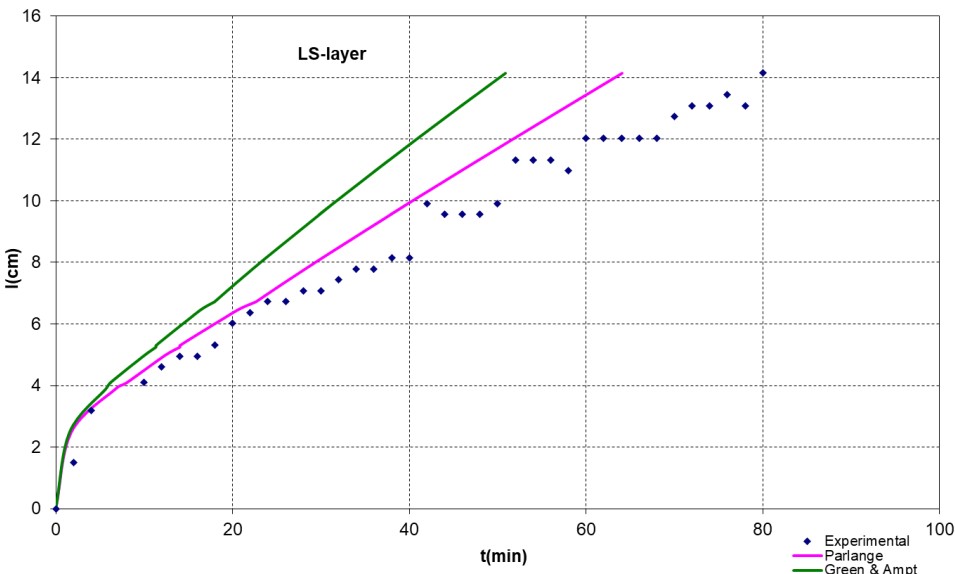

**Figure 7.** Approximation of the infiltration models with the experimental points for the upper layer (LS-layer).

### 3.2. Pressure Head–Soil Moisture Research

#### 3.2.1. First Drainage

The changes in soil moisture with time were captured by the TDR sensors and are presented in Figure 8a,b for both layers. Figure 9a,b show the pressure head changes over time, as they were captured by the P.T and recorded by the logger for the LS-layer and S-layer, respectively.

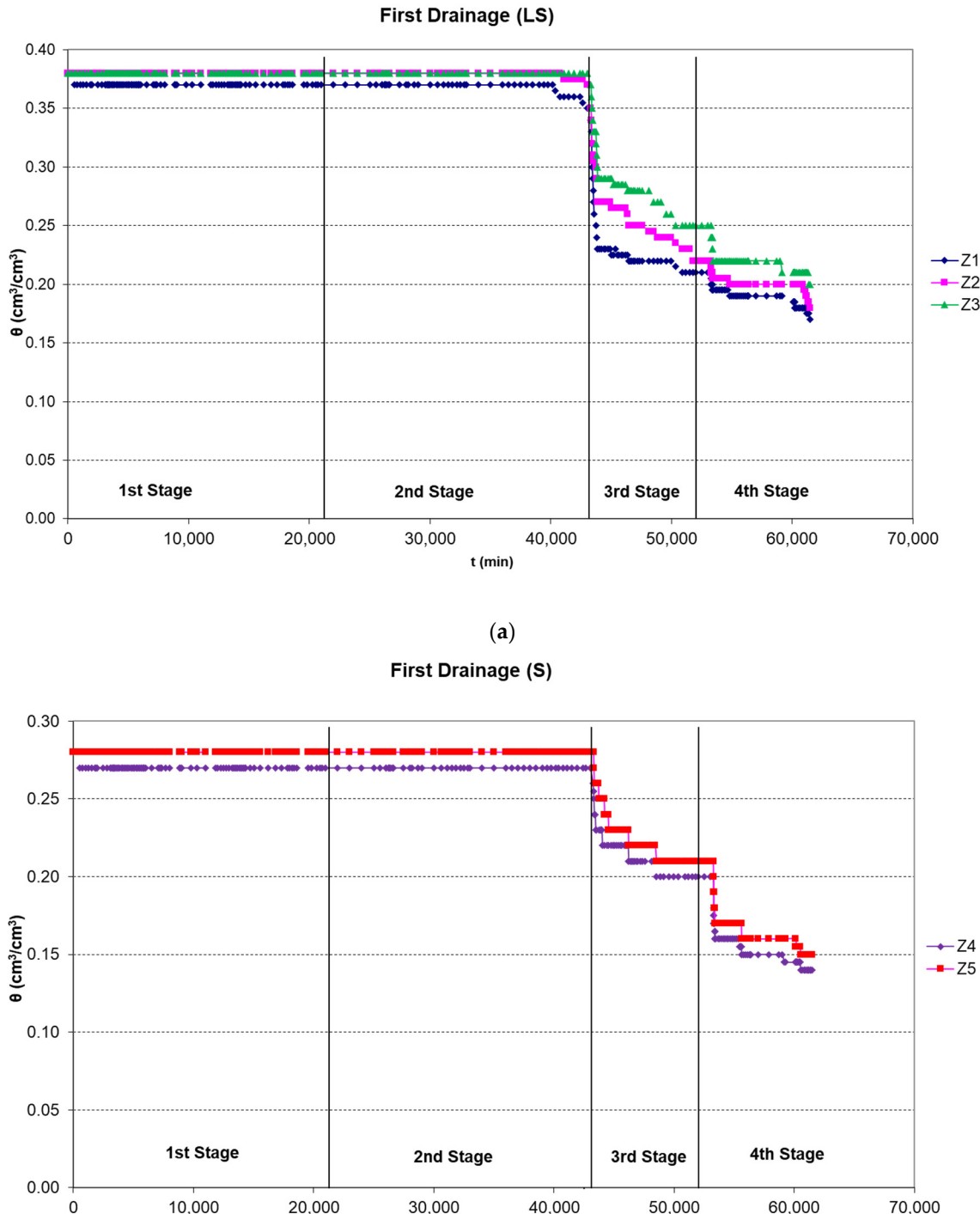

**Figure 8.** Soil moisture vs. time for the LS-layer (**a**) and S-layer (**b**) at first drainage.

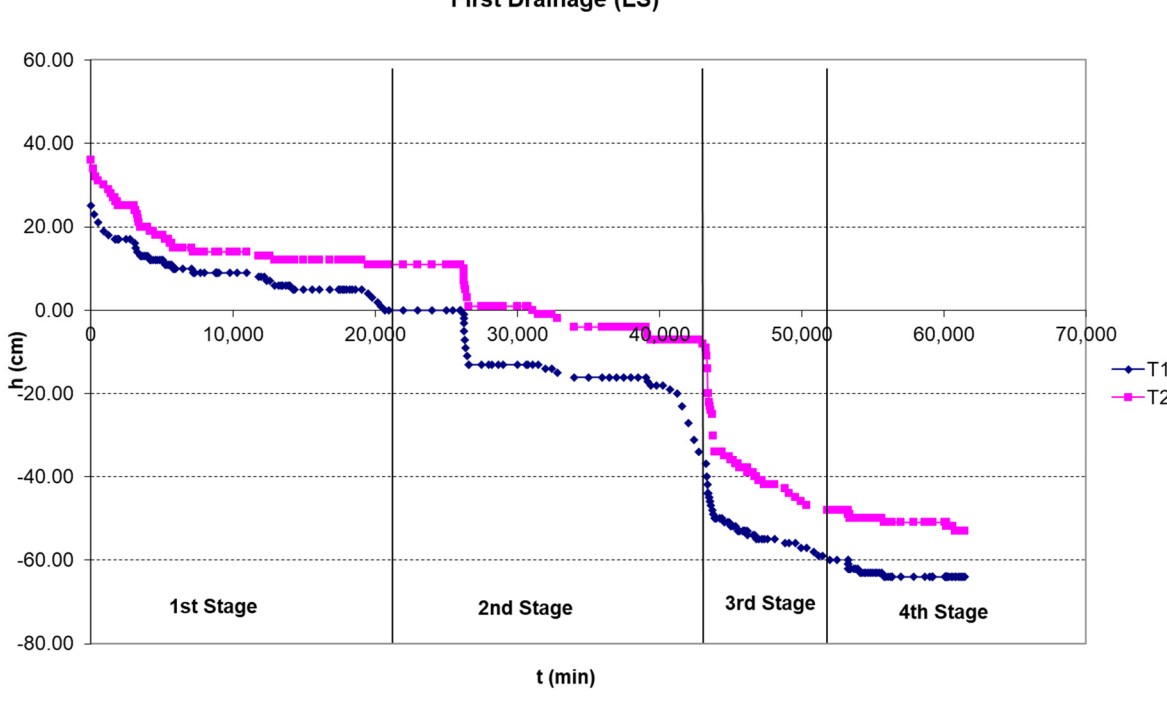

(**a**)

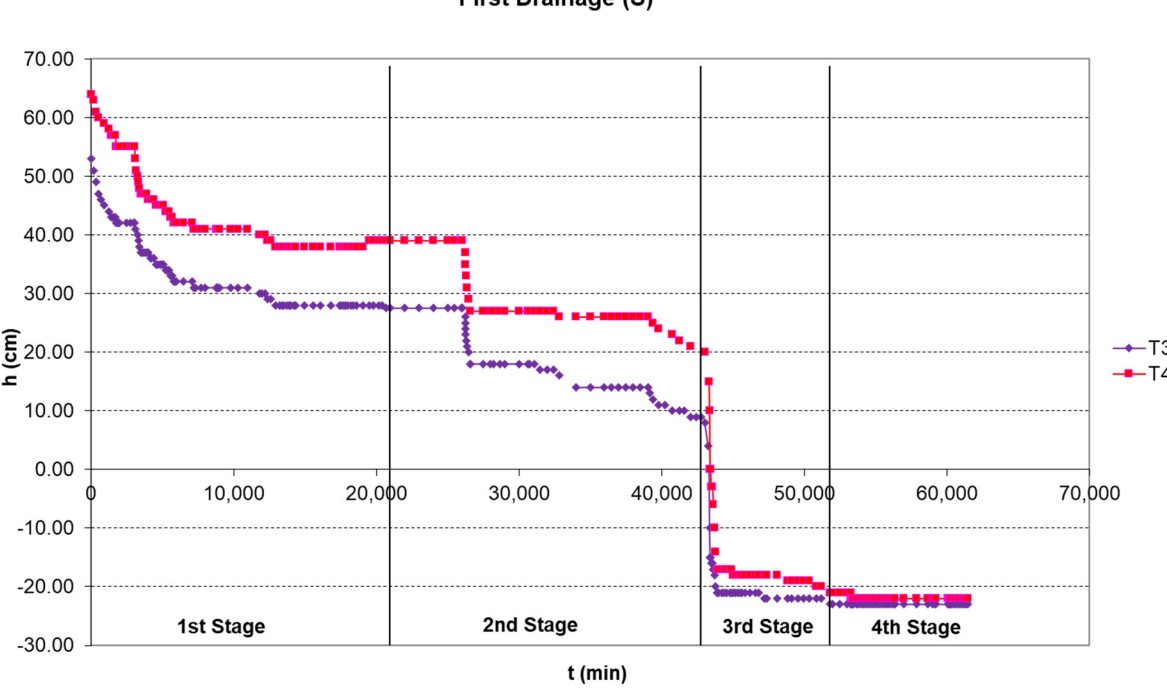

(**b**)

**Figure 9.** Pressure head vs. time for LS-layer (**a**) and S-layer (**b**) at first drainage.

3.2.2. Second Imbibition

From the experimental data, the soil moisture vs. time, along with pressure head vs. time, were extracted (Figures 10a,b and 11a,b, respectively).

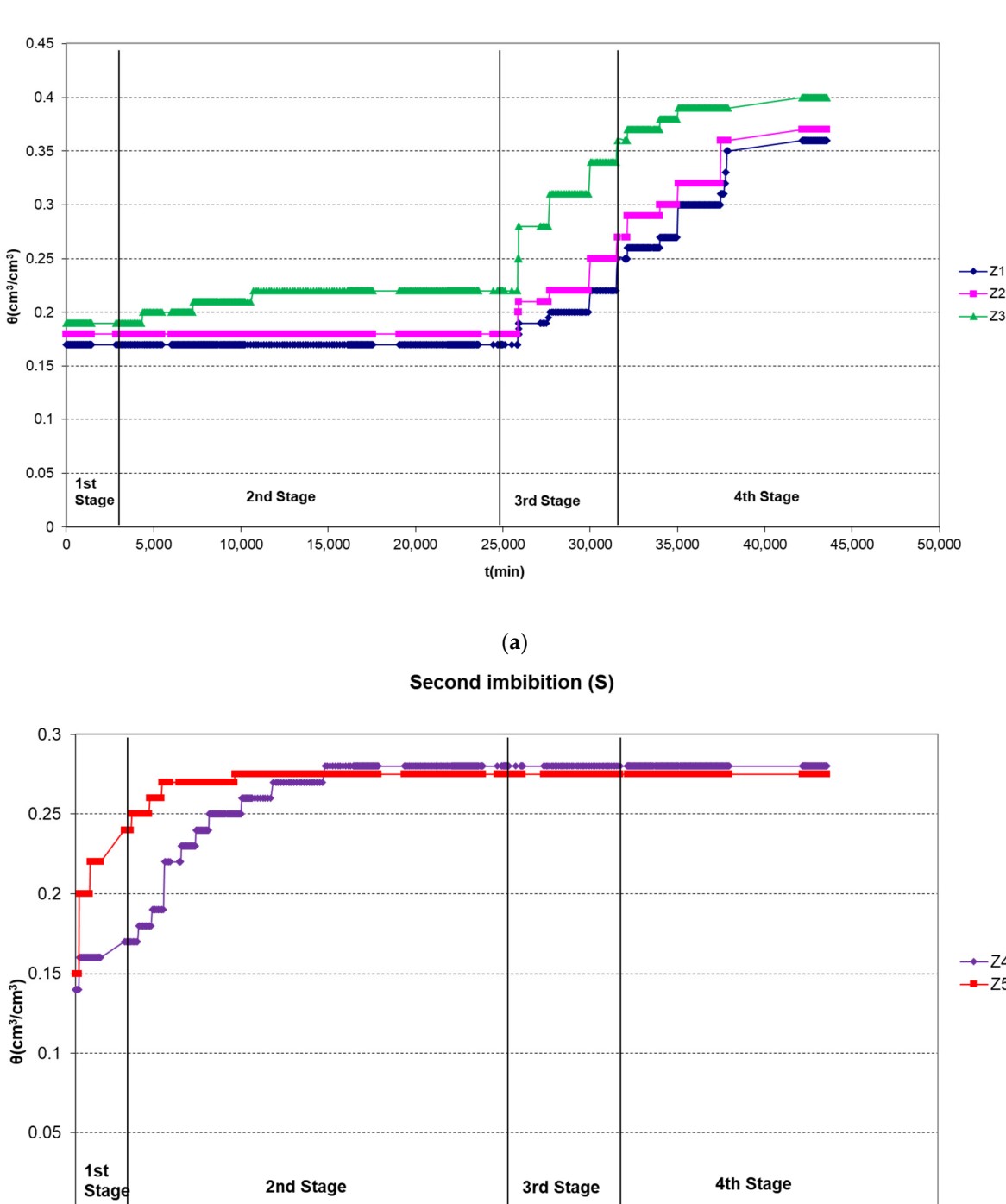

**Figure 10.** Soil moisture vs. time for the LS-layer (**a**) and S-layer (**b**) at the second imbibition.

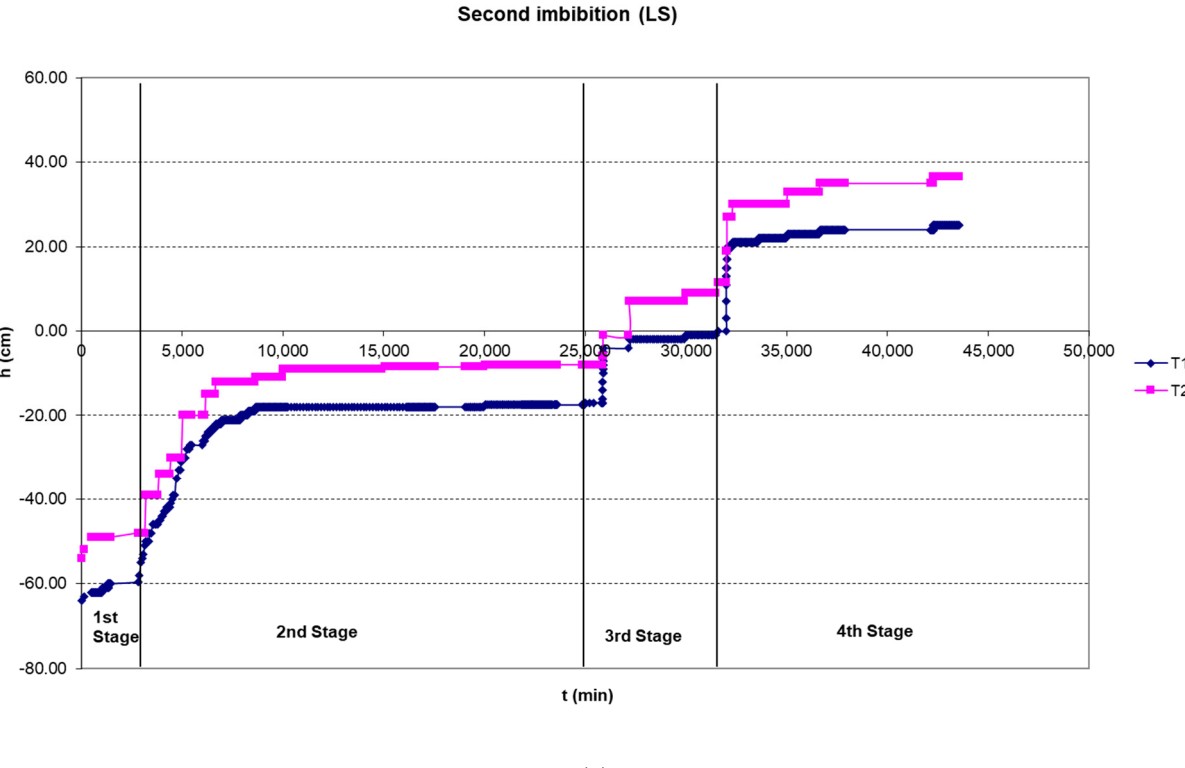

(**a**)

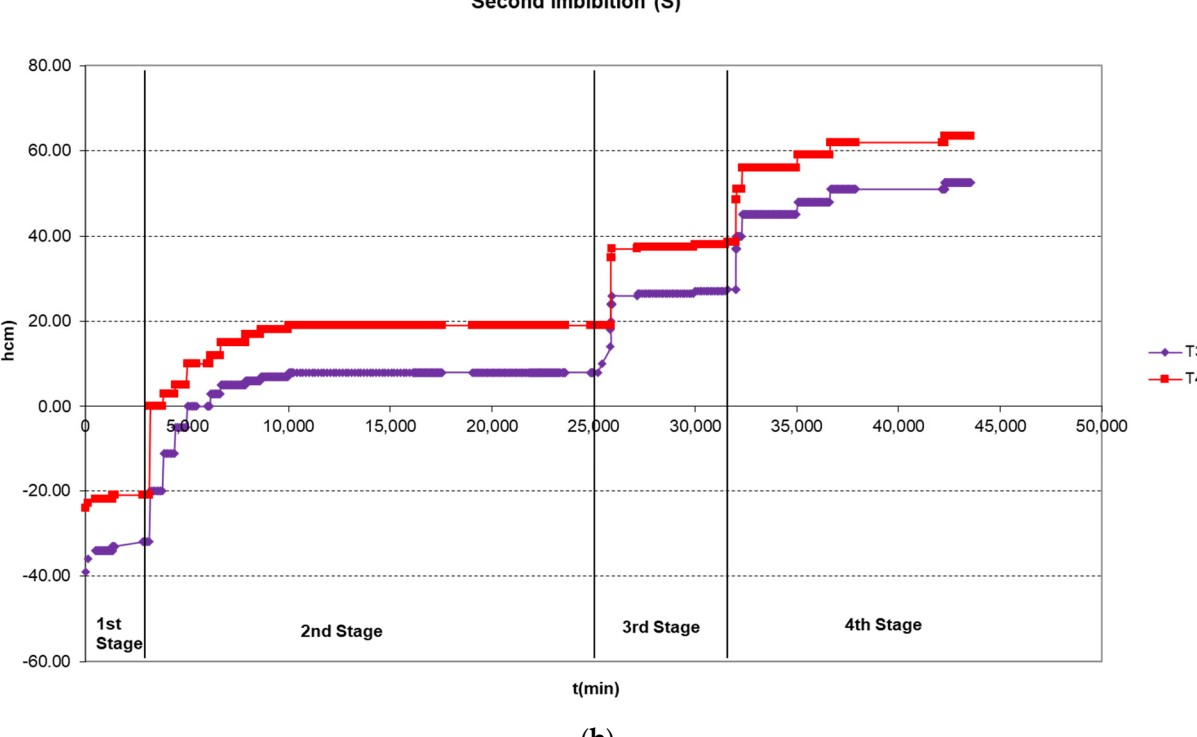

(**b**)

**Figure 11.** Pressure head vs. time for the LS-layer (**a**) and S-layer (**b**) at the second imbibition.

### 3.2.3. Characteristic Curves of the Two Layers

The soil water characteristic curves (SWCC) of both layers were obtained from the soil moisture and pressure head versus time curves and are presented in Figure 12a,b.

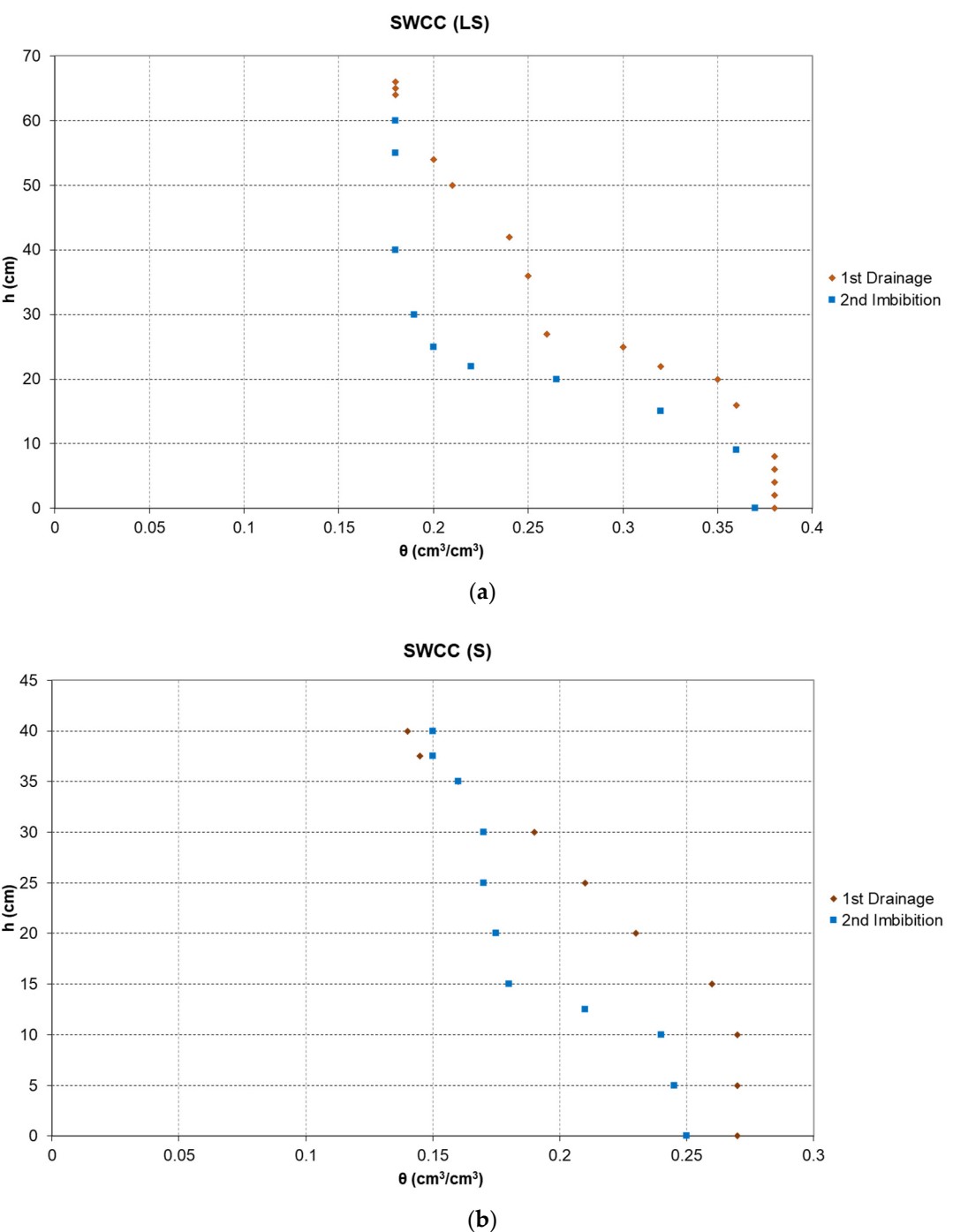

**Figure 12.** SWCC LS-layer (**a**) and S-layer (**b**).

The estimation of the hydraulic parameters of van Genuchten's model (11) for the S-layer obtained from RETC software (Code for Quantifying the Hydraulic Functions of Unsaturated Soils, U.S. Salinity Laboratory, USDA, ARS, Riverside, California) and are reported in Table 1:

**Table 1.** Hydraulic parameters of van Genuchten's model (RETC).

| Hydraulic Parameter | $\theta_s$ | $\theta_r$ | $\alpha$ | $n$ | $m$ |
|---|---|---|---|---|---|
| RETC value | 0.24937 | 0.16 | 0.0797 | 6.47629 | 0.845591 |

Diffusivity $D(\theta)$ was obtained from Equations (4)–(7) and simulated in Figure 13. The fitting Equation is given below:

$$D(\theta) = 5.7e^{18,000(\theta-0.182)^3} \tag{13}$$

Integral (3) was used to calculate sorptivity, but due to the impossibility of the analytical solution of the Integral (14):

$$S^2 = 2\int_{\theta_i}^{\theta_s} \theta D(\theta)d\theta = 2\int_{\theta_i}^{\theta_s} \theta\,5.7e^{18,000(\theta-0.182)^3}\,d\theta \tag{14}$$

We proceeded to a graphical solution. For this purpose, we divided the fitting curve of $D(\theta)$ into three parts, found their fitting Equations and calculated the separated integrals. Thus, $S^2$ was found as a sum of the separated integrals (Table 2).

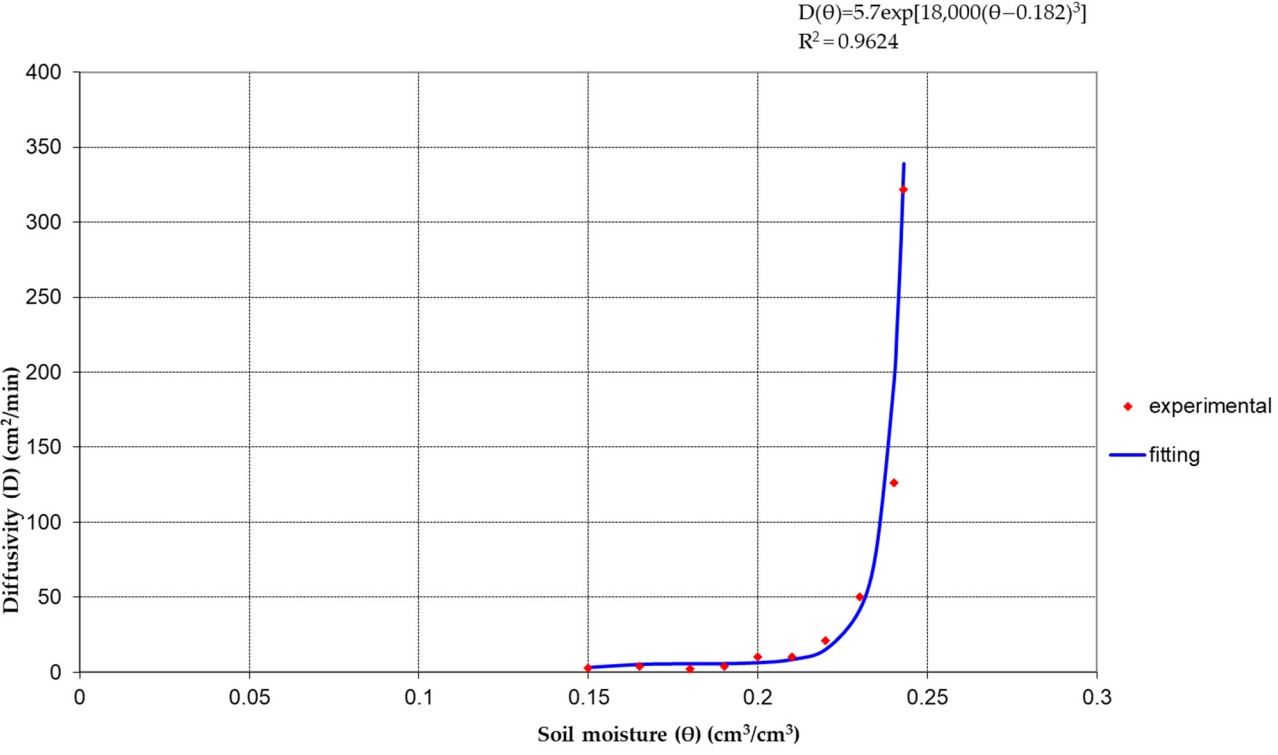

**Figure 13.** Diffusivity vs. soil moisture.

**Table 2.** Graphical solution of Integral (13) and estimation of sorptivity of the S-layer.

| | | |
|---|---|---|
| 1st part: ($\theta$: 0.15–0.18) | $D_1 = 84.632\theta - 9.5346$ ($R^2 = 1$) | $S_1^2 = 2\int(84.632\theta^2 - 9.5346\theta)\,d\theta = 0.4420$ |
| 2nd part: ($\theta$: 0.18–0.21) | $D_2 = 92.099\theta - 10.879$ ($R^2 = 1$) | $S_2^2 = 2\int(92.099\theta^2 - 10.879\theta)\,d\theta = 0.0830$ |
| 3rd part: ($\theta$: 0.21–0.24) | $D_3 = 6085.9\theta - 1269.6$ ($R^2 = 1$) | $S_3^2 = 2\int(6085.9\theta^2 - 1269.6\theta)\,d\theta = 1.3737$ |
| $S^2 = 1.5012$ cm·min$^{-0.5}$ | | |
| $S = 1.2252$ cm min$^{-0.5}$ | | |

Hence, the sorptivity ratio for the two layers is given below:

$$\frac{S_S}{S_{LS}} = \frac{1.2252}{1.0607} = 1.155$$

indicating that sorptivity values raise as hydraulic conductivity raises (as the soil texture gets more light) [21,22,24,27,44].

## 4. Discussion

In Figures 6 and 7, it is observed that at the initial times of water movement, the gradient of the curve is considerable, as, in the beginning, capillary forces are the dominant forces. In other words, capillary forces are greater than gravity, but later, gravity is the main force acting [40]. Also, in the beginning, infiltration is intense, while afterward, the gradient $dI/dt$, which expresses the infiltration rate, decreases. The decrease in infiltration rate over time is due to several factors, such as the deformation of the soil structure, soil pore clogging, and the possibility of trapped air into the soil pores, but mainly due to the reduction of the hydraulic gradient [21,30,34–36,45]. Specifically, during the vertical movement of water into completely dry soil, the ponding conditions at the soil surface cause immediate saturation at the soil surface. Hence, the hydraulic gradient at the soil layer very close to the surface is steep, but over time, as water moves towards greater depths, the gradient decreases [46]. The continuous reduction of the hydraulic gradient near the soil surface results in a continuous reduction of the infiltration rate, which eventually stabilizes at a certain value. This constant value of the infiltration rate is practically equal or tends toward the saturated hydraulic conductivity ($K_s$) [21,30,31,47,48]. The values of the measured saturated hydraulic conductivity tend to the gradient $dI/dt$ at late infiltration times (relative mse = 0.002), indicating good validation of the experimental process. Infiltration experimental data of the upper layer were approximated with the GA and P models, and results showed relative mse 0.11 and 0.04, respectively, in agreement with previous infiltration experiments under ponded conditions [21,23].

Figure 6 shows a comparison of the volumetric values of the infiltrated water with those obtained by integrating the moisture profiles indicating good agreement and validating the experimental process. Thus, integrating the soil water content vs. depth leads to a good approximation of the infiltration process [20]. Hence, in case of missing $I(t)$ data, $\theta(z)$ curves can lead to the prediction of cumulative infiltrate, indicating that the soil moisture profiles curve is a useful tool to simulate infiltration over time.

Regarding soil water content during the first two stages of the 1st drainage, no significant changes in soil moisture of the LS-layer were observed, but during the third stage, we observed a decrease in moisture, as seen in Figure 8a. During the third stage, where the drainage bottle was placed −21 cm from the 4th CC (T4), a gradual decrease of moisture was observed, while the bigger decrease was detected gradually at the first sensor, then at the second and the smallest decrease at the third sensor, as expected, because the first sensor was placed 16.5 cm from the second, 26.5 cm from the third sensor and 72.5 cm from the drainage bottle. Thus, at the current stage, the gradient $d\theta/dt$ is significant, as expected [40,44,46]. During the fourth stage of drainage, a decrease in soil moisture at all three sensors of the upper layer was observed but lesser than that during the third stage. At the end of the stage, the experimental water content values were stabilized; specifically, the first sensor showed the lowest soil moisture value (0.17 cm$^3$/cm$^3$) as expected, while the second sensor was stable at 0.18 cm$^3$/cm$^3$ and the third sensor at 0.20 cm$^3$/cm$^3$. Figure 8b shows the reduction of water content at the S-layer during the four stages of the 1st drainage. It is noticed that at the first two stages of the drainage, we observed absolutely no change in the layer's water content. This was expected during the first two stages, and the drainage bottle was placed above the two TDR sensors (Z4 and Z5); thus, no drainage could occur. During the third stage, when the drainage container was located −21 cm from the fourth CC, therefore below the two soil moisture sensors, a significant water content decrease was observed. Moreover, the $d\theta/dt$ gradient was steep at the beginning of the stage, indicating low holding capacity [45,46]. During the fourth stage, when the drainage container was placed −26 cm from the fourth CC, we observed a further decrease and, afterward, the stabilization of water content. The Z4 sensor stabilized at 0.14 cm$^3$/cm$^3$ and the Z5 at 0.15 cm$^3$/cm$^3$.

Unlike water content, tension, as a more sensitive variable [41,42,44] decreasing from the first stage of the first drainage (Figure 9a). Thus, during the first two stages, we observed significant changes in tension, while in the third stage, we observed a significant tension decrease, too, as the drainage bottle was set to −48 cm from the second CC (T2) and −58.5 cm from the first CC (T1), At the beginning of the third stage the gradient of the $h(t)$ curve was significant and gradually decreased at the end of the stage and finally stabilized at the 4th stage, where equilibrium occurred, and gradient tended toward zero. The final value of tension recorded by the first tension sensor (T1) was −64 cm, while for the second sensor (T2) was −53 cm, which is justified by the fact that the T2 sensor was placed at −11.5 cm lower than the T1 sensor, indicating that the stabilized values of tension tend to the external loads applied [22].

Figure 9b shows that during the first two stages of drainage, the decrease of tension at the S-layer is not as significant as at the LS-layer. The fact can be explained by the position of the drainage bottle in relation to the CC. In the first stage, the drainage bottle was placed at the same level as the first CC (T1). During the second stage, it was placed at the interface, and during the third stage, it was set at −21 cm from the fourth CC (T4). Thus, a significant decrease was observed for the two lower sensors (T3 and T4). Hence, at the beginning of the third stage, the gradient of the $h(t)$ curve is steep, while later, the tension stabilizes, as can be seen in Figure 9b. Finally, at the fourth (final) stage, we see that equilibrium has been achieved, and sensors T3 and T4 stabilize at −23 cm and −22 cm, respectively, almost equal to the external loads applied [41,42,49].

During the first stage of the second imbibition, soil moisture showed no significant changes at the LS layer, as expected, due to the position of the Mariotte bottle (−21 cm from the fourth CC). Thus, water couldn't reach the TDR sensors. During the second stage, the Mariotte bottle was placed at a height of −8 cm from the Z3 sensor and small increases in soil moisture were detected at the Z3 sensor. This could be explained by the action of the capillary phenomenon because although the water level was at a lower level than the sensor, the sensor detected increasing water content [40,44,49,50]. The other two moisture sensors showed stabilized water content, probably because the capillary length was shorter than the locations of the sensors [22,40]. During the third stage, the Mariotte bottle was placed at the height of the first CC (−25 cm from the soil surface), and a significant increase was observed at all sensors of the upper layer. A greater increase in soil moisture was detected by the Z3 sensor, as expected, due to its location (−33.5 cm from the soil surface). During the final (fourth) stage, the Mariotte bottle was placed at the height of the soil surface, and a further increase of soil moisture was observed, along with stabilization at values close to $\theta_s$.

At the S-layer, the two relative sensors showed quite small increases in soil moisture during the first stage (the Mariotte bottle was placed −21 cm from the fourth CC), while their values increased significantly during the second stage (the Mariotte bottle was placed at the interface), which was expected, as the location of the Mariotte bottle justifies such an increase.

As already mentioned, tension, in contrast to soil moisture, is more sensitive and showed a remarkable increase from the beginning of imbibition. This fact is justified because tension is instantly affected by the applied load, while significant changes in the soil are observed when a great load (either positive or negative) is applied due to soil water capacity [22,40,51]. During the first stage, we observed a small increase of pressure at sensors T1 and T2 at the LS layer, which, in the second stage, increased significantly. From the middle of the second stage to its end, we observed the stabilization of the tension of both sensors. During the third and fourth stages, the pressure increased further, and at the end of the fourth stage, it stabilized at 25 cm and 36.5 cm, respectively, indicating that the tension values tend to the applied external loads [22,40].

At the S-layer, tension increase was observed from the very first stage of the second imbibition since the pressure is instantly affected by the applied pressure head. The increase of tension values became greater during the second stage, where the Mariotte bottle was

placed at the interface of the two soil layers. During the next two stages, the values of tension were increased further for both tension sensors (T3, T4), and at the end of the 4th stage, equilibrium was achieved. At the beginning of the third and fourth stages of imbibition, the derivative dh/dt is steep, indicating the instant increase of tension into the soil's pores. Gradually, dh/dt decreases and finally tends to zero at equilibrium [2,22,40].

From Figures 8–11, SWCC $h(\theta)$ was extracted for each layer (Figure 12a,b). The hydraulic hysteresis phenomenon is significant at both layers indicating that the pressure head-soil moisture function is not a 1-1 function [22,46]. The S-layer shows a greater hysteresis loop than the LS-layer, which can be explained by the nature, structure and physical properties of the sandy soil, which has low holding capacity along with lower values of saturated soil moisture ($\theta_s$) and residual soil moisture ($\theta_r$). During the draining and wetting circles of the soil column, we obtained differences between the soil moisture values of the two layers. On the contrary, the pressure head changes show similar behavior in both soils. This is in agreement with the theory of hydraulic hysteresis [49,52–54], where it is indicated that the area under the drainage curve expresses the work required to lead to the complete drainage of a unit volume of saturated soil. The same goes for imbibition. Hence, the difference between the above areas is equal to what is called hydraulic hysteresis work [$\frac{W}{\gamma} = \int_0^{\theta_s} h(\theta)d\theta$], where $\gamma$ is the specific weight of water. Hydraulic hysteresis work is greater as the soil texture becomes lighter [22,52,54].

Regarding the estimation of sorptivity, results showed that the sorptivity ratio was 1.155, indicating that the S-layer has greater sorptivity. This is in accordance with theory and previous research, mentioning that lighter-textured soils with greater hydraulic conductivity at saturation show greater sorptivity [21,23,26,27,29–31]. Under ponded conditions, for $m$ = 1, Equation (1) can lead to the estimation of sorptivity. We can also lead to sorptivity estimation through calculations of hydraulic capacity and diffusivity using Equations (3)–(7). Due to the impossibility of the analytical solution of the Integral (13), we proceeded to a graphical solution of the integral by dividing it into three parts and approximating each part with linear Equations. Summing the results of the three integrals led to the estimation of the sorptivity of the S-layer. Sorptivity is a crucial hydraulic parameter that has a great impact on water movement through porous media. As it is a component of the flow process, it needs to be incorporated in applications where the adsorption or desorption of water from a porous media is occurring [22,24,55–59]. The results of the current study suggest deep research into the vadose zone of the soils under research, using methods to estimate crucial hydraulic parameters that are involved in the soil-water motion, constituting useful tools for further investigation of runoff, aquifer horizon recharge, rational water management and water saving.

## 5. Conclusions

The objective of this research was to determine the hydraulic parameters and properties of a layered soil column. Experiments were accomplished in the laboratory for a layered soil (Loamy Sand over Sand) using TDR probes and ceramic capsules connected to pressure transducers. Cumulative volumes of water versus time were figured using experimental data. Soil moisture versus time and moisture profiles were also figured using TDR data. Integration of moisture profiles $\theta(z)$ led to a good approximation of the infiltration process; hence, soil moisture profiles can simulate infiltration over time. Green and Ampt and Parlange infiltration models were used to predict infiltration for the upper layer with good results. Thus, using early time data is a quick and easy way to predict water movement under ponded conditions. Drainage and imbibition experimental circles were used to extract the Soil Water Characteristic Curves of each layer, indicating the hydraulic hysteresis effect during the drainage and imbibition processes. Simulation of Soil Water Characteristic Curves led to van Genuchten's hydraulic parameters assessment, and further mathematical modeling of water motion led to the estimation of specific water capacity and diffusivity. To avoid the impossibility of an analytical solution of the integral that provides sorptivity, a graphical solution was used. The results indicate that the methods

used in the current study are useful for the simulation of vertical water motion through the under-research layered soils.

**Supplementary Materials:** The following are available online at https://www.mdpi.com/article/10.3390/w15020221/s1, Figure S1: Calibration curves of the CC-PT system.

**Author Contributions:** Conceptualization, A.A. and I.C.; methodology, A.A. and I.C.; software, A.A.; validation, A.A., I.B. and I.C.; formal analysis, A.A.; investigation, A.A. and I.C.; resources, A.A., I.B. and I.C.; data curation, A.A., I.B. and I.C.; writing—original draft preparation, A.A.; writing—review and editing, A.A. and I.B.; visualization, A.A. and I.B.; supervision, A.A.; project administration, A.A. All authors have read and agreed to the published version of the manuscript.

**Funding:** This research received no external funding.

**Data Availability Statement:** Data available on request due to restrictions, e.g., privacy or ethics. The data presented in this study are available on reasonable request from the corresponding author. The data are not publicly available due to privacy and copyright reasons.

**Conflicts of Interest:** The authors declare no conflict of interest.

## Abbreviations

| | |
|---|---|
| a | Hydraulic parameter in Van Genuchten's model |
| $\theta$ | soil moisture |
| $\theta_i$ | initial water content |
| $\theta_s$ | soil moisture at saturation |
| $C$ | Specific water capacity |
| CC | Ceramic capsule(s) |
| $D$ | Diffusivity |
| $h$ | Suction |
| $H_f$ | Suction at the wet front |
| $H_0$ | Pressure head at the soil surface |
| $I$ | Cumulative infiltration |
| GA | Green and Ampt |
| $K_i$ | Hydraulic conductivity |
| $K_s$ | Hydraulic conductivity at saturation |
| LS-layer | Loamy sand layer |
| $m$ | Hydraulic parameter in Van Genuchten's model |
| $n$ | Hydraulic parameter in Van Genuchten's model |
| P | Parlange |
| PT | Pressure transducer(s) |
| $S$ | Sorptivity |
| S-layer | Sand layer |
| $S_m$ | A series of coefficients in Philip's Equation |
| SWCC | Soil water characteristic curve(s) |
| $t$ | time |
| TDR | Time domain reflectometry |

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
