# Peer review of "Hydrodynamics of the Vadose Zone of a Layered Soil Column"

_water, doi:10.3390/w15020221_

Round 1
Reviewer 1 Report
This paper needs more effort from authors to improve the quality of figures numbers and clearance, it is very poor.
I am afraid about the methodology of this manuscript for the experimental method used in this study, so the authors must calibrate this method using anther experimental test and apply to the field test to be applicable method.
I needs to see the updated version
Author Response
Thank you for your comments, please see the attached file.

Reviewer 2 Report
Dear authors, I had the pleasure of revising your paper entitled:" Hydrodynamics of the vadose zone of a layered soil column" whose purpose is to suggest a method to estimate the main hydraulic parameters that are involved in the soil-water infiltration, runoff, aquifer horizon recharge, water management, and water-saving, through a series of laboratory experiments. The investigation of unsaturated media is very complex, and the authors provide a substantial introductory background with the various studies carried out by other authors. However, I believe the manuscript has several weaknesses, which I have indicated as specific comments in the attached PDF file. Furthermore, I think the work is more the connotations of a technical report, helpful in describing a methodology and an instrumental apparatus possibly applicable to other layered porous media. The English language is often difficult to follow, making the manuscript difficult to understand in some parts. Please, make the changes suggested in the PDF in order to proceed with a subsequent possible revision from my side.
Best.

Author Response

(The authors gave the same response as above.)

Round 2
Reviewer 2 Report
Dear authors, I have re-read your manuscript and I cannot refrain from saying that it has been much improved since the revisions effort was made from your side. Just a point of clarification about your comment: "Would you please explain why there is “short-circuit with the results” at L119 of the PDF file provided (or L158 of the revised manuscript)? This line just describes how the soil layers were placed into the column…". I was referring to line 118: "The two soil samples had a different texture and hydraulic properties". For me, this sentence should be removed cause the texture has been evaluated by the granulometric tests and the hydraulic properties have been determined in detail by this research so I think it is much more suitable to move this sentence to the result section and avoid this phrase at the beginning of materials and methods section. In brief, to say that the two types of soil used have different textures and different hydraulic characteristics is trivial in this session, but it is less trivial in the light of your numerical results, emphasizing if only partially, the results of your manuscript. Please consider changing that phrase. I have no more comments on the work.
Best.
Author Response

(The authors gave the same response as above.)
